# Research on extremely short construction period of engineering project based on labor balance under resource tolerance

**Junlong Peng, Mengyao Wang** **\*, Chao Peng, Ke Hu**

College of Transportation Engineering, Changsha University of Science and Technology, Changsha, China

\* 1732057408@qq.com

## Abstract

Under the condition of resource tolerance, engineering construction projects face the problem of labor force balance in the working face. Notably, a deviation occurs between the distribution and certain demand of the labor force in the limited working face, which affects the realization of an extremely short construction period. To address this problem, we first introduced the stochastic coefficient of labor force equilibrium to measure the degree of labor balance. Second, a labor force equilibrium model with the realization goal of an extremely short construction period was established. Then, the standard particle swarm optimization (PSO) algorithm was improved from two perspectives to solve the proposed model. The update equation was rounded to solve practical project problems, and a dynamic variable inertia weight was adopted to ensure the PSO algorithm accuracy and convergence speed. Finally, through case analysis, we determined the extremely short construction period and best labor force distribution scheme. Moreover, the case results revealed that the established model is simple, operable and practical and that the proposed algorithm achieves a high search accuracy and efficiency in the model solution process. Overall, under the condition of resource tolerance, this study provides scientific and effective references for managers to realize an extremely short construction period.

**Data Availability Statement:** All relevant data are within the paper.

**Funding:** The authors gratefully acknowledge the funding and support provided by the Natural

## 1. Introduction

The construction period of engineering projects has always been considered an important research topic in the construction industry in China and abroad. In domestic engineering projects, the problem of the construction period has remained of great concern. In recent years, major emergencies have frequently occurred in China. Temporary rescue sites, road and bridge restoration, emergency hospitals and other projects have required each builder to rapidly respond to achieve loss and damage minimization [1–4]. Due to the incident urgency, taking Huoshenshan and Leishenshan Hospitals under COVID-19-related constraints as an example [5–7], China raised the efforts of the whole society to provide a large number of resources to ensure rapid high-quality construction within an extremely short period. The world was amazed by the construction speed of these two emergency hospitals. However,

Science Foundation of Hunan Province, China (No: 2021JJ30746), (No: 2015JJ32004).

**Competing interests:** The authors have declared that no competing interests exist.

under the condition of a large quantity of aggregated human, financial and material resources, i.e., resource tolerance, compression of the project duration to the limit and realization of an extremely short construction period have become notable research issues.

At present, scholars have mainly focused on resource constraints and the shortest construction period in engineering project management [8–12]. In particular, under resource constraints, scholars have investigated methods to reasonably arrange the start time of each activity based on satisfying the logical relationship among project activities, thereby minimizing the project duration. Additionally, the above has been demonstrated to be a nondeterministic polynomial time (NP)-hard problem [13,14], and the research in this field largely includes the following two aspects:

1. In regard to the shortest construction period of a single project under resource constraints, Zhang et al. [15] established an optimization model aimed at project construction period minimization and effectively solved the problem via particle swarm optimization (PSO) based on priority and permutation. Peng et al. [16] further proposed the particle representation method based on priority permutation. Compared to the former method, the latter approach was verified to solve the problem more effectively. Vahid et al. [17] formulated a construction sequence plan with the realization goal of the shortest construction period based on building information modeling (BIM), developed computer programs with a genetic algorithm, and generated a stable construction schedule. Liu et al. [18] considered the factors of the material supply, cost constraints and various labor modes, established a model with the minimum construction project duration as the primary goal, and effectively solved the abovementioned problem. Xie et al. [19] focused on the constraints of the prefabricated component supply for prefabricated buildings, conducted in-depth research on the corresponding scheduling problem, effectively and reasonably distributed resources and reduced the completion time.

2. In terms of the shortest construction period of multiple projects under resource constraints, Marimuthu et al. [20] examined, summarized and compared optimization modeling methods. Suresh [21] and Goncalves [22] applied a genetic algorithm to shorten the project duration and improve the utilization rate of resources of the project group through resource allocation. Mohamed et al. [23] developed a multiobjective scheduling optimization model, which could enable construction enterprises to solve resource conflicts under the condition of multiple project priorities and the distribution of limited resources. Wang et al. [24] evaluated multiple projects based on priority, proposed a schedule model with the shortest weighted construction period of multiple projects as the goal, and solved the proposed model with an adaptive PSO algorithm. Hauder et al. [25], based on the minimum multiproject construction period, proposed two goals: activity balance and resource balance. This approach was demonstrated to be applicable by solving the mixed-integer programming-based constraint model constructed in a large project.

All the above studies have provided an important reference and suggestions for the realization of an extremely short construction period of a given project under resource constraints. In contrast, few scholars have performed research on the achievement of an extremely short construction period under the condition of resource tolerance. However, against the background of COVID-19 and innovation-driven development in the 14th Five-Year Plan [26], it is necessary to thoroughly study the realization of an extremely short construction period of a project from the new perspective of resource tolerance. In the research process, it has been found that even under the condition of resource tolerance, there remain many factors influencing the realization of an extremely short construction period in terms of the engineering

quantity [27,28], management [29–31], technology [31,32], and environment [33,34]. This study only focuses on the factor of labor force balance under working face limitations.

Under the condition of resource tolerance, due to the limitation of the working face, we can face the following two situations affecting construction period compression: when the distribution of the labor force in each working face is lower than a certain demand, we cannot increase the construction speed nor minimize the construction period to the highest degree. In addition, many resources (human, financial and material resources) can be wasted. When the distribution of the labor force in each working face is higher than a certain demand, the increase in labor force is not directly proportional to the construction speed. In other words, workers can decrease their work efficiency through working face reduction, thereby affecting the realization of an extremely short construction period. Therefore, under the condition of resource tolerance, it is necessary to perform in-depth research on the realization of an extremely short construction period of a project considering the important influencing factor of labor force balance in the limited working face. We should continuously optimize and adjust the labor force distribution in the limited working face, reduce the deviation between the labor force distribution and demand, balance the labor force distribution and demand, and finally realize an extremely short construction period of the engineering project.

To solve this problem scientifically and effectively, this paper first introduces the stochastic coefficient of labor force equilibrium, which effectively optimizes and adjusts the labor force by measuring the degree of labor force equilibrium in the limited working face. Next, the labor force is balanced by reducing the deviation between the labor force distribution and demand. Then, a labor force equilibrium model with the realization goal of an extremely short construction period is established. Based on a labor force balance in the limited working face, an extremely short construction period of the engineering project can be realized. Finally, the paper improves the standard PSO algorithm from two perspectives: the update equation is rounded to solve practical project problems, and a dynamic inertia weight is adopted to ensure the PSO accuracy and convergence speed. Subsequently, the improved PSO algorithm is employed to solve the research model, and the corresponding extremely short construction period and best labor force distribution scheme are determined. This study can provide theoretical support for project managers to realize an extremely short construction period of engineering projects under the condition of resource tolerance.

## 2. Problem description and research hypothesis

### 2.1 Problem description

It is assumed that a project comprises a set of $V = [V_0, V_1, V_2, ..., V_n, V_{n+1}]$ activities, where activities $V_0$ and $V_{n+1}$ are dummies (no consumption of time and resources, respectively) and denote the initial and final project activities, respectively. The duration and start time of activity $V_i$ ($i = 1, 2, ..., n$)$\epsilon V$ are denoted as $d_i$ and $s_i$, respectively. The project duration $T$ is determined by the start time $s_{n+1}$ of activities $V_{n+1}$, and we set the project start time to 0, i.e., $s_0 = 0$. The engineering quantity of activity $V_i \epsilon V$ is denoted as $C_i$, and the total labor allocation, total labor demand and labor output quota of activity $V_i \epsilon V$ are denoted as $R_i$, $Q_i$, and $E_i$, respectively.

### 2.2 Research hypothesis

To facilitate analysis, the following hypotheses are established:

1. Under the condition of resource tolerance, this paper achieves an extremely short construction period with quality assurance.

2. The duration of each activity is not rounded to preserve the accuracy of the determination of an extremely short construction period.

3. The operation process of each activity cannot be interrupted, and the quantities of each activity remain fixed.

4. Under the condition of resource tolerance, the labor force distribution in the working face of each activity is independent, and there occurs no delay or failure to conduct an activity according to the normal plan due to an insufficient labor force.

5. The impact on the construction period is the same when the labor force distribution in the working face of each activity is higher than or lower than the same unit of the labor force demand.

## 3. Research model

### 3.1 Stochastic coefficient of labor force equilibrium $K$

In this study, the goal of realizing an extremely short construction period of the project is reached under the premise of a labor force balance in the limited working face of each activity. Hence, to measure the degree of labor balance in the working face, we introduced the stochastic coefficient of labor force equilibrium $K$. Notably, the imbalance in the labor force can be divided into two cases in this paper: $R_i > Q_i$ and $R_i < Q_i$. Therefore, the expression of the stochastic coefficient of labor force equilibrium ($K_i$) is as follows:

$$K_i = \begin{cases} R_i/Q_i & R_i > Q_i \\ R_i/Q_i \ or \ Q_i/R_i & R_i = Q_i \\ Q_i/R_i & R_i < Q_i \end{cases} \quad \in [1, \ z_{max}] \tag{1}$$

Where $K_i$ denotes the stochastic coefficient of labor force equilibrium in the working face of each activity. When the value of $K$ approaches 1, the labor force becomes increasingly balanced. For $K_i = 1$ ($R_i = Q_i$), the labor force is completely balanced and reaches the ideal state. $z_{max}$ is a constant greater than 1 and represents the maximum acceptable value of the stochastic coefficient of labor force equilibrium. In the limited working face, given the safe distance and working efficiency, $Z_{max} = 1.5$.

### 3.2 Labor force equilibrium model

Based on the above comprehensive analysis, this paper finally realizes an extremely short construction period of the project by continuously optimizing and adjusting the labor force distribution in the limited working face, reducing the deviation between the distribution and certain demand of the labor force and constantly balancing the labor force. Therefore, the labor force equilibrium model can be formulated as follows:

$$T = min \ s_{n+1} = \sum_{V_i \in CP} d_i \tag{2}$$

$$\text{Subject to } minR_i \leq R_i \leq maxR_i \tag{3}$$

$$s_i \geq 0, \ d_i \geq 0 \tag{4}$$

$$C_i \geq 0, \ R_i \geq 0, \ Q_i \geq 0 \tag{5}$$

Eq (2) expresses the objective function of this model, where $CP$ is the critical path of the project, which comprises the key activities. Eq (3) defines the constraint of the labor force distribution, which controls the distribution of the labor force and cannot exceed the scope during optimization to ensure meaningful optimization. Eqs (4) and (5) are nonnegative constraints of the time and labor force, respectively, in the engineering project.

The exhaustive calculation steps of the objective function are as follows:

Step 1: According to the maximum acceptable value of the stochastic coefficient of labor force equilibrium $z_{max}$ in the limited working face, the distribution conforming to the workforce distribution scheme should be limited between two known constant maximum ($maxR_i$) and minimum ($minR_i$) values, and other conditions are not considered. Consequently, the total labor force demand of activity $V_i$ can be calculated as follows:

$$Q_i = \left\lceil \left[ \frac{1}{2} \left( minR_i \cdot z_{max} + maxR_i / z_{max} \right) \right] \right\rceil$$
$$R_i \in [minR_i, \quad maxR_i] \tag{6}$$

Step 2: $K_i$ is calculated according to Eqs (1) and (5).

Step 3: Combining the above steps, the duration of each activity can be calculated as follows:

$$d_{i=} \begin{cases} \dfrac{C_i}{R_i E_i} & R_i \leq Q_i \\[2ex] \dfrac{C_i}{Q_i E_i} \cdot K_i & R_i > Q_i \end{cases} \tag{7}$$

Step 4: $CP$ is determined by applying the critical path method to obtain construction period $s_{n+1}$.

## 4. Solution method

The PSO algorithm was first proposed by Kennedy and Eberhat in 1995 as a bionic evolutionary algorithm [35]. The PSO algorithm dictates that particles fly at a specific speed in the search space, and the flight speed and position of each particle are continuously optimized and updated through information sharing between particles. Consequently, particles gradually reach the optimal position and obtain the best fitness value [36]. As an intelligent algorithm for global optimization of complex problems based on populations, the PSO algorithm has been widely applied to solve complex optimization problems in many fields. It has been verified that this algorithm provides the advantages of simplicity, easy implementation and good robustness [37–40]. The problem in this study is addressed under the condition of resource tolerance. By solving the labor force balance in each limited working face in the project, an extremely short construction period of the project can be realized, which is essentially a duration optimization problem. Therefore, based on the PSO algorithm, this paper improves the evolution equation and inertia weight parameters of this method, designs a corresponding algorithm according to the research model, and finally effectively solves the problem.

### 4.1 Coding scheme

Under the condition of resource tolerance, this paper established a labor force equilibrium model aimed at the determination of an extremely short construction period. The purpose of this practice is to continuously adjust the labor force distribution in the limited working face

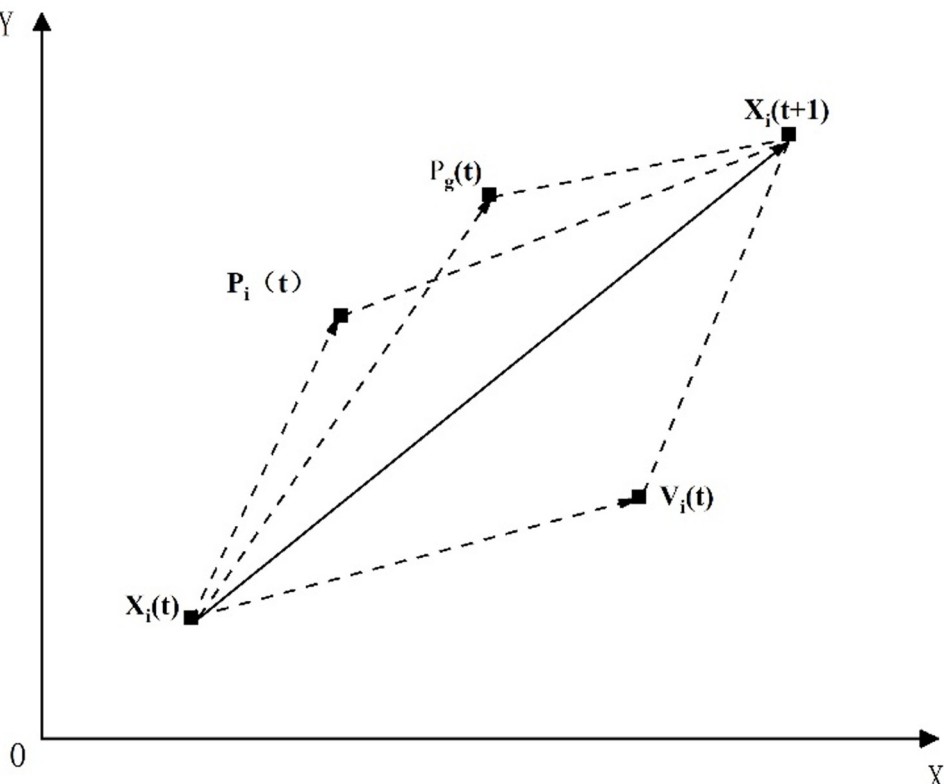

**Fig 1. Mechanism of particle movement in space.**

within a known labor force distribution range, reduce the deviation between the distribution quantity and certain demand of the labor force, continuously balance the labor force and finally achieve the realization goal of an extremely short construction period of the project.

Based on this principle, we assumed that there exist M particles in the N-dimensional feasible solution search space of the objective problem, where N denotes the number of jobs in the problem and M denotes the size of the particle swarm (the number of particles). The current speed of particle $i$ is expressed as $V_i(t) = (v_{i1}, v_{i2}, ..., v_{iN})$. The current position of particle $i$ is expressed as $X_i(t) = (x_{i1}, x_{i2}, ..., x_{iN})$, which represents a feasible solution of the objective problem, where the value of $x_{ij}$ ($i = 1, 2, ..., M; j = 1, 2, ..., N$) corresponds to the actual labor force distribution. The speed $V_i(t+1)$ of particle $i$ at the next time step depends on the current speed $V_i(t)$, its best position $P_i(t)$ and the global best position $P_g(t)$. Each particle moves to the next position $X_i(t+1)$ through speed updating. The position movement mechanism of the above particle in space is shown in Fig 1. $[x_{j\ min}, x_{j\ max}]$ is the range of activity of the particles in spatial dimension $j$, where $x_{j\ min}$ denotes the minimum labor force distribution for activity $j$, and $x_{j\ max}$ denotes the maximum labor distribution for activity $j$. The particles are continuously optimized and updated in the search space to gradually reach the best particle position. In particular, the best labor force distribution scheme in each working face of the project is consequently obtained. At this time, the fitness value represents the optimized extremely short construction period.

## 4.2 Evaluation function

The evaluation function is also regarded as the fitness value function, which is calculated to evaluate the particle position. In other words, this function is employed to evaluate the

advantages and disadvantages of the feasible problem solution, and an iterative update process is thus implemented until the optimal solution is obtained. The objective function of the model established in this study is the determination of an extremely short construction duration of the project. Therefore, considering this goal, the model objective function Eq (2) is selected as the evaluation function. Generally, when $T$ is small, the particle position is excellent. Notably, the better the labor distribution scheme is, the more balanced the labor force in the limited working face.

### 4.3 Improvement of the evolution equation of PSO

This paper improved the PSO equation from the perspective of practical engineering projects. If the number of laborers in a given project is required to be an integer, the actual distribution of the labor force in the working face of each activity corresponding to $x_{ij}$ should therefore be an integer. Therefore, the adjusted evolution equation is expressed as follows:

$$v_{ij}(t+1) = int\big(\omega v_{ij}(t)\big) + int\big(c_1 r_1[p_{ij}(t) - x_{ij}(t)]\big) \tag{8}$$

$$+int\big(c_2 r_2[p_{gj}(t) - x_{ij}(t)]\big)$$

$$x_{ij}(t+1) = v_{ij}(t+1) + x_{ij}(t) \tag{9}$$

Where $\omega$ is the inertia weight value, $c_1$ and $c_2$ are the two speed factors of self-cognitive learning and social learning, respectively, $r_1$ and $r_2$ are two random numbers, generally in the interval of [0,1], $t = 1, 2,...G$ is the number of iterations, and $G$ is the maximum number of iterations. In addition, Eqs (8) and (9) are adopted to update the speed and position, respectively.

### 4.4 Inertia weight ω

**4.4.1 Dynamic variable inertia weight.**   Generally, the inertia weight of the standard PSO algorithm is a fixed value, which is likely to yield premature particles, resulting in the local optimization phenomenon [41–43]. Therefore, we improved the accuracy and convergence speed of the algorithm by using a dynamic inertia weight. The dynamic variable inertia weight in this study was proposed by Ren [44] by introducing and defining the change rate of the

**Table 1. Algorithm performance test results.**

| Function name | Dimension | Variable range | Strategy | Optimum fitness value | Success rate (%) |
|---|---|---|---|---|---|
| Griewank | 30 | [−600,600] | PSO | 9.8531 | 10 |
| | | | Improved PSO | 2.94 | 45 |
| Rastrigin | 30 | [-5.12,5.12] | PSO | 5.376 | 30 |
| | | | Improved PSO | 4.106 | 65 |
| Rosenbrock | 30 | [−30,30] | PSO | 6.924 | 45 |
| | | | Improved PSO | 4.816 | 50 |

focusing distance. The dynamic variable inertia weight can be expressed as follows:

$$\omega = \begin{cases} (\alpha_1 + r/2)\ln|k| & |k| > 1 \\ \alpha_1\alpha_2 + r/2 & 0.05 \leq |k| \leq 1 \\ (\alpha_1 + r/2)/|\ln|k|| & |k| \leq 0.05 \end{cases} \tag{10}$$

$$k = \frac{MaxDist - MeanDist}{MaxDist} \tag{11}$$

$$MeanDist = \frac{\sum_{i=1}^{M}\sqrt{\sum_{n=1}^{N}(p_g - p_i)^2}}{M} \tag{12}$$

$$MaxDist = max\sqrt{\sum_{n=1}^{N}(p_g - p_i)^2} \tag{13}$$

Where $k$ is the change rate of the focusing distance, $MaxDist$ is the maximum focusing distance, $MeanDist$ is the average focusing distance, and $r$ is a random number uniformly distributed within the interval of [0,1]. Commonly, $\alpha_1 = 0.3$ and $\alpha_2 = 0.2$.

Numerical analysis [44] has verified that the proposed adaptive PSO algorithm with a variable inertia weight obtains satisfactory results in terms of the solution accuracy and convergence speed.

**4.4.2 Performance test of the improved PSO algorithm with a dynamic variable inertia weight.** To verify the effectiveness of the improved PSO algorithm with a dynamic variable inertia weight proposed in this study, three test functions were compared to the standard PSO algorithm in the simulation environment of MATLAB R2017b. The test function equations are expressed as follows:

(1) Griewank:  $\quad f_1 = \sum_{i=1}^{n}\frac{x_i^2}{4000} - \prod_{i=1}^{n}\cos\left(\frac{x_i}{\sqrt{i}}\right) + 1$

(2) Rastrigin:  $\quad f_2 = 10n + \sum_{i=1}^{n}x_i^2 - 10\cos(2\pi x_i)$

(3) Rosenbrock:  $\quad f_3 = \sum_{i=1}^{n}[100(x_{i+1} - x_i^2)^2 + (x_i - 1)^2]$

In the above two algorithms, the population number is 30, the maximum number of iterations is 1000, and the other parameter settings remain the same. Both algorithms are independently run 30 times, and the test results are listed in Table 1. The optimum fitness value of the three functions based on the improved PSO algorithm with a dynamic variable inertia weight is the smallest, and the success rate is obviously higher than that of the standard PSO

algorithm, which demonstrates that the algorithm proposed in this paper achieves a good optimization ability. Therefore, we applied the proposed algorithm in follow-up research.

### 4.5 Algorithm steps for model solution

In conclusion, algorithm design of the labor equilibrium model to realize an extremely short construction period is achieved as follows:

1. Preparatory work before algorithm implementation: the objective function and constraints are input, the data for each case task are read, and the algorithm parameters are set;

2. Initialization and calculation of the fitness value of each particle: the speed and position of all particles are initialized according to the specific conditions of the project to produce an initial matrix;

3. Iterative evolutionary update: $\omega$ is determined based on Eqs (10) and (11), the velocity and position of all particles in the population are updated according to Eqs (8) and (9), respectively, and the fitness value after each iteration is calculated;

4. Evaluation of particles: after each evolution iteration, the fitness value of each particle is calculated and compared to obtain $p_i$ and $p_g$, and the next iteration is entered;

5. Iteration termination condition setting: when the number of iterations meets the maximum number of iterations $G$, the algorithm process is terminated, and the final output result comprises $T(p_gbest)$, $R_i$, $d_i$, and $K_i$. Otherwise, the algorithm returns to step (3), and the iteration process is continued.

6. End.

   The specific solution process is shown in Fig 2.

## 5. Case study

### 5.1 Case construction

Currently, there is no database related to the problem in this study. However, to illustrate the practical operability of the proposed model and the accuracy and efficiency of the solution algorithm, this paper designed a suitable simulation instance, which involves a highway engineering project with 20 real activities and a contract period of 350 days. The name and related parameters of each activity are listed in Table 2. Among these parameters, those named after bulldozers and scrapers indicate that the construction content of these activities mainly entails mechanical operation. In addition, since the units of measurement of each activity differ and most activities contain multiple specific construction contents, to facilitate analysis, the quantities of each activity are abstracted as comprehensive quantities without units of measurement, and the corresponding labor output quota is a comprehensive labor output quota. According to tight front and tight back relationships between the various activities, a network plan is obtained, as shown in Fig 3.

### 5.2 Simulation results

In the MATLAB R2017b environment, we imported relevant project data and coded the model solution process based on the proposed algorithm. During encoding, the initial parameters were set as follows: the size of the population $M = 50$; the dimension of the search space $N = 20$; the initial inertia weights $\omega_{max} = 0.95$ $and$ $\omega_{min} = 0.25$; the learning coefficient $c_1 = c_2 =$

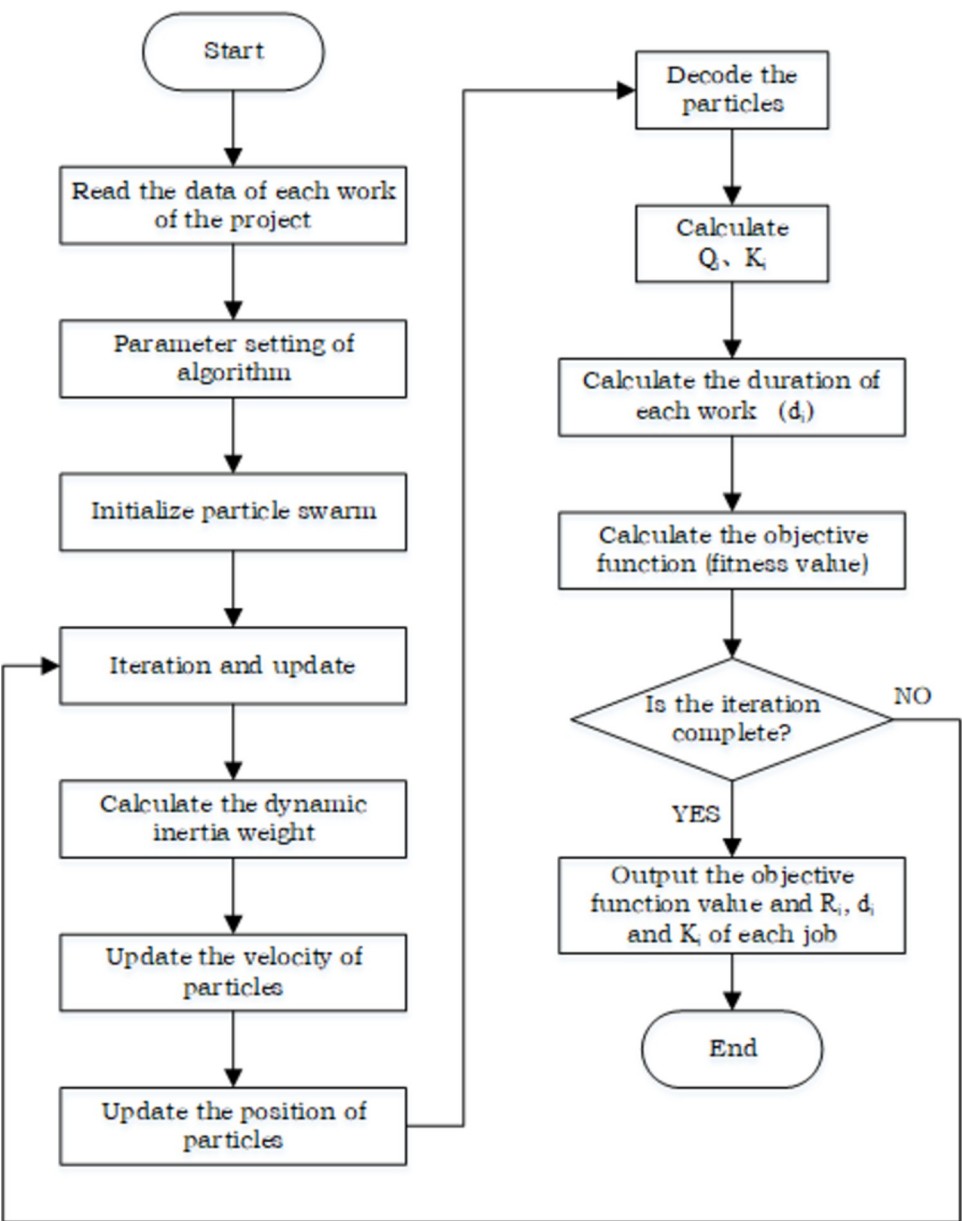

**Fig 2. Improved PSO solution flowchart.**

2; and the maximum number of iterations $G = 200$. The algorithm was operated 50 times, and the iterative output results are listed in Table 3.

Table 3 reveals that the extremely short construction duration of the project reaches 253.26 days. Compared to the contract period, the construction period is 27.64% shorter. Moreover, the specific duration of each activity is provided in Table 3 and intuitively shown in Fig 4. Consequently, the critical path of this project is ①→②→③→④→⑧→⑩→⑫→⑮→⑯→⑰→⑱. Furthermore, a bar chart of the schedule corresponding to the obtained extremely short construction period of the project was generated, as shown in Fig 5.

**Table 2. Relevant parameters of each activity.**

| Serial number | Activity name | Code | Comprehensive quantities | Comprehensive labor output quota (/day) | Labor distribution | |
|---|---|---|---|---|---|---|
| | | | | | Minimum value | Maximum value |
| **1** | **Preparation** | **1–2** | **1500** | **6** | **20** | **40** |
| 2 | Bulldozer I | 2–3 | 71000 | 500 | 3 | 5 |
| 3 | Excavation and filling earthwork | 2–10 | 235000 | 300 | 7 | 9 |
| 4 | Bulldozer II | 3–4 | 134000 | 500 | 4 | 6 |
| 5 | Slab culvert wall | 3–5 | 4230 | 3 | 45 | 65 |
| 6 | Tube sheet channel | 5–7 | 3000 | 3 | 25 | 40 |
| 7 | Circular pipe culvert | 3–6 | 2500 | 4 | 25 | 40 |
| 8 | Retaining wall | 3–10 | 7260 | 3 | 50 | 70 |
| 9 | Scraper operation | 4–8 | 105000 | 400 | 5 | 8 |
| 10 | Rapid stream trough | 6–9 | 5650 | 8 | 30 | 50 |
| 11 | Aqueduct | 9–10 | 4000 | 8 | 30 | 50 |
| 12 | Interval processing | 7–10 | 3200 | 5 | 25 | 40 |
| 13 | Bed course I | 8–10 | 13400 | 12 | 50 | 70 |
| 14 | Bed course II | 10–11 | 12960 | 12 | 50 | 70 |
| 15 | Base course I | 10–12 | 13200 | 10 | 50 | 70 |
| 16 | Base course II | 13–14 | 12760 | 10 | 66 | 82 |
| 17 | Surface course I | 12–15 | 13000 | 5 | 65 | 82 |
| 18 | Surface course II | 16–17 | 12500 | 5 | 65 | 82 |
| 19 | Clearing I | 15–18 | 13000 | 34 | 56 | 78 |
| 20 | Clearing II | 17–18 | 12500 | 34 | 70 | 90 |

Table 3 and Fig 6 show that the actual labor force distribution of each activity was obtained. Actually, the result represents the best labor force distribution scheme of the project. Fig 7 shows the distribution of the stochastic coefficient of labor force equilibrium ($K_i$) for each

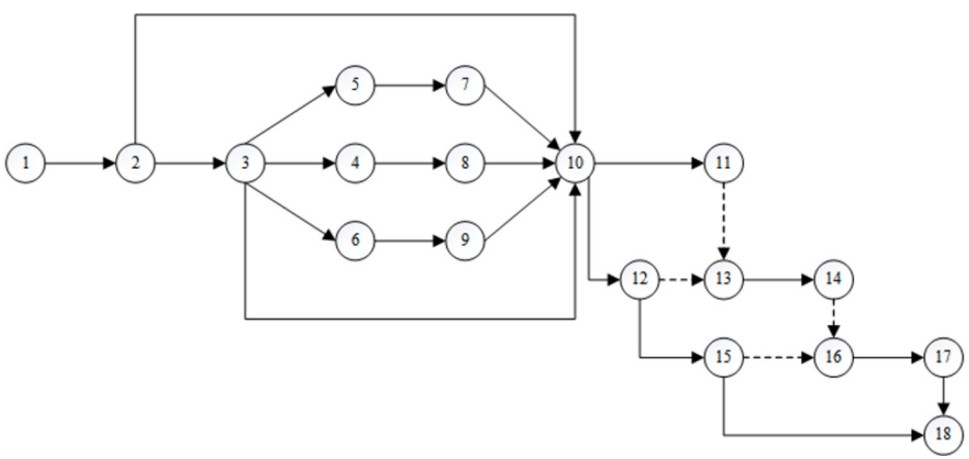

**Fig 3. Project double-generation network plan.**

**Table 3. Calculation output results.**

| Serial number | Optimal labor force distribution | Duration of each activity (day) | Disequilibrium coefficient ($K_i$) | Equilibrium deviation $\Delta K_i = K_i - 1$ |
|---|---|---|---|---|
| 1 | 28 | 8.93 | 1.036 | 0.036 |
| 2 | 4 | 35.50 | 1.000 | 0.000 |
| 3 | 8 | 97.92 | 1.125 | 0.125 |
| 4 | 5 | 53.60 | 1.250 | 0.250 |
| 5 | 54 | 26.11 | 1.037 | 0.037 |
| 6 | 31 | 32.26 | 1.065 | 0.065 |
| 7 | 31 | 20.16 | 1.065 | 0.065 |
| 8 | 63 | 39.68 | 1.033 | 0.033 |
| 9 | 6 | 43.75 | 1.167 | 0.167 |
| 10 | 43 | 17.66 | 1.075 | 0.075 |
| 11 | 39 | 12.82 | 1.026 | 0.026 |
| 12 | 31 | 20.65 | 1.065 | 0.065 |
| 13 | 61 | 18.31 | 1.000 | 0.000 |
| 14 | 63 | 17.71 | 1.033 | 0.033 |
| 15 | 61 | 21.64 | 1.000 | 0.000 |
| 16 | 75 | 17.01 | 1.027 | 0.027 |
| 17 | 76 | 34.21 | 1.013 | 0.013 |
| 18 | 76 | 32.89 | 1.013 | 0.013 |
| 19 | 74 | 5.62 | 1.088 | 0.088 |
| 20 | 83 | 4.43 | 1.000 | 0.000 |
| Construction period $T(p_g best)$ | | 253.26 | | |

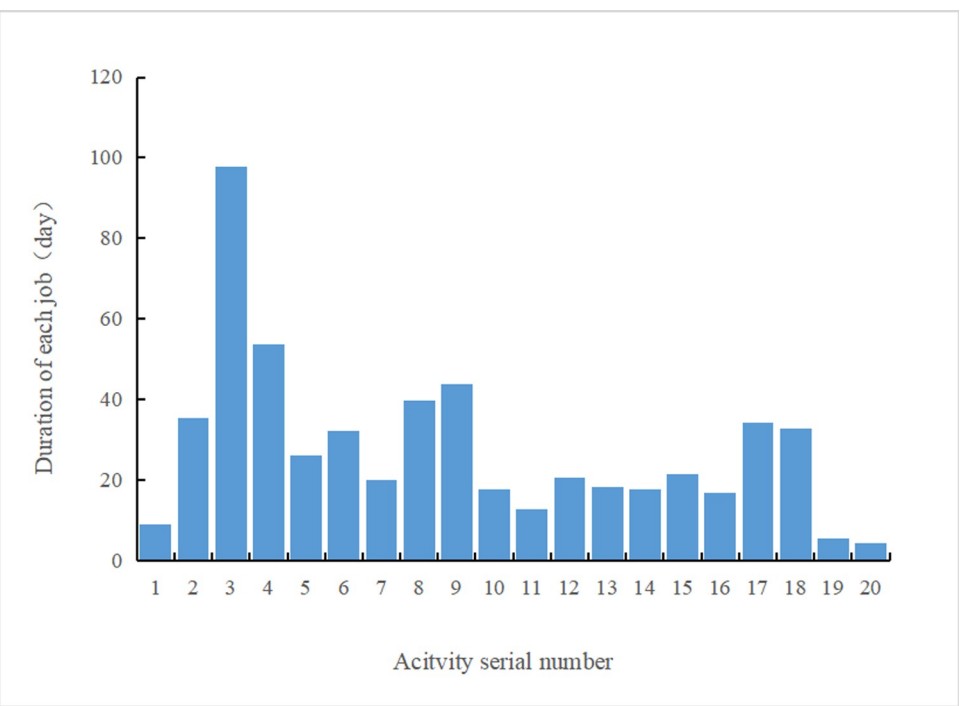

**Fig 4. Duration of each activity.**

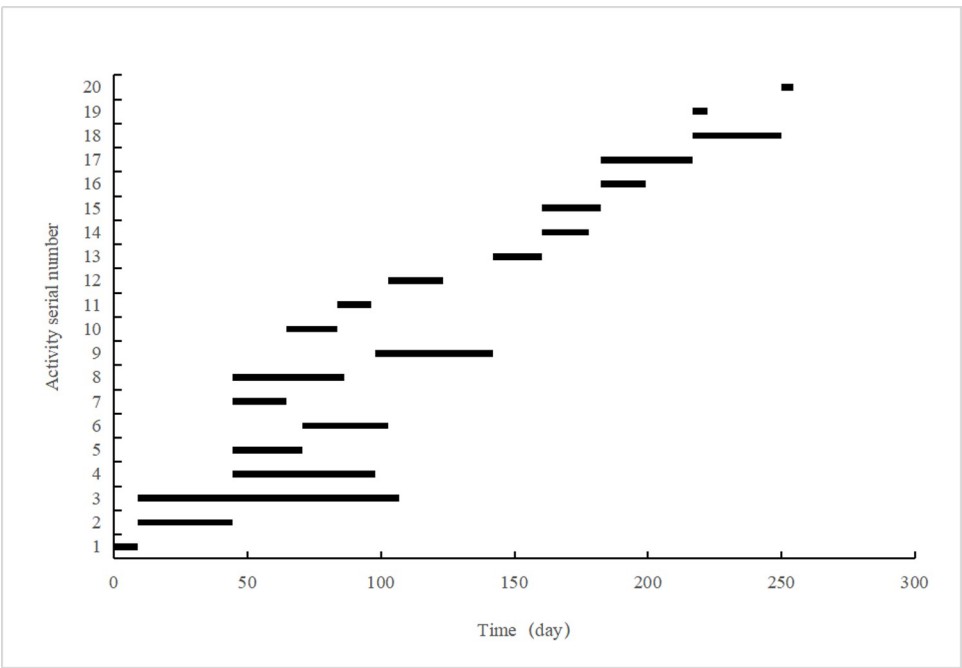

**Fig 5. Bar chart of project schedule.**

activity. Except for the activities involving human–machine cooperation, the equilibrium deviation $\Delta K_i$ is no greater than 0.100, indicating that an extremely short construction period is realized based on the balance among the various working labor forces. Thus, the obtained scheme achieves a suitable reliability. Moreover, the solution process gradually converges in

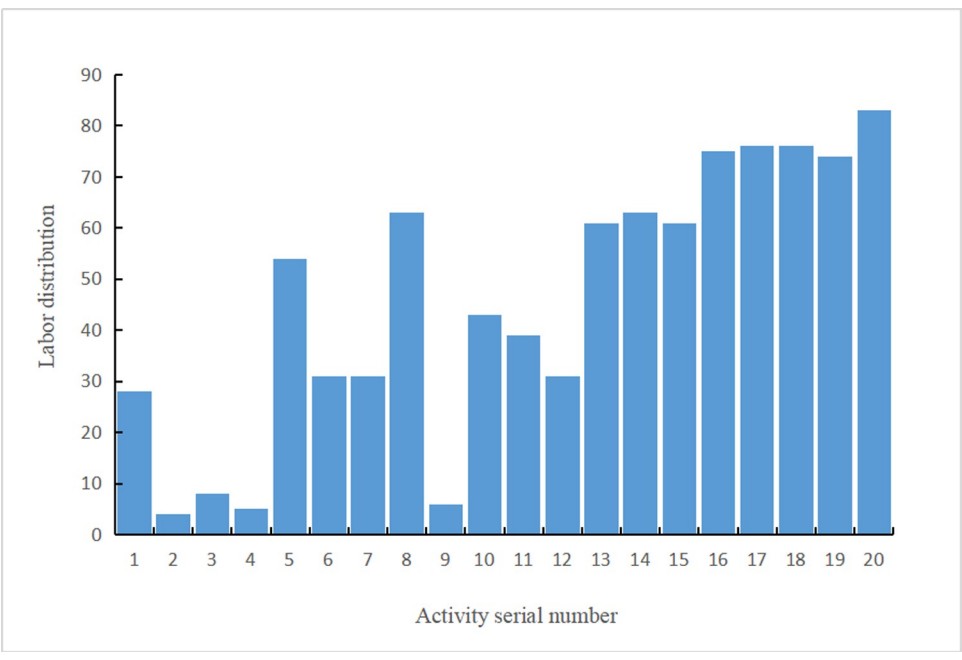

**Fig 6. Labor force distribution of each activity.**

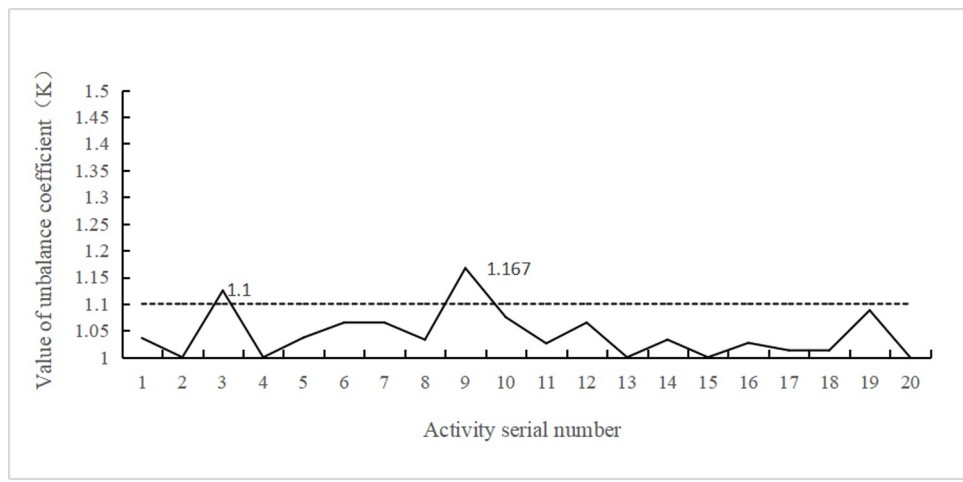

**Fig 7. Value of $K_i$.**

this study. After approximately 25 generations, convergence is accomplished to yield the optimal solution, which verifies the feasibility of the model and algorithm to solve practical problems of engineering projects (the convergence process is described in the next section).

## 5.3 Comparison of the results and calculation efficiency

To further verify the superiority of the improved PSO algorithm in this paper, we compared the simulation results between the standard PSO algorithm and proposed improved PSO algorithm. Here, the parameters of these two algorithms were set to be the same, and the designed case was again simulated. Consequently, a performance comparison table of these two algorithms was constructed, as summarized in Table 4. As such, a comparison of the evolution curves of these two algorithms is shown in Fig 8.

According to Table 4 and Fig 8, we found that the results and efficiency of the improved PSO algorithm are better than those of the standard algorithm. In terms of the target function value, the minimum time limit of the improved PSO algorithm is 253.26 days, which is shorter than the time limit of 256.19 days obtained with the standard PSO algorithm. In terms of the convergence speed, the proposed algorithm with a dynamic variable inertia weight converged onto the optimal solution in 25 generations, which is 5.2 times faster than the convergence realization of the standard PSO algorithm. Therefore, the improved PSO algorithm proposed in this paper achieves a preferable accuracy and efficiency in regard to the actual case.

## 6. Conclusion

Under the condition of resource tolerance, based on a labor force balance in the limited working face, an extremely short construction period of the project can be realized. This study demonstrates that the stochastic coefficient of labor force equilibrium introduced can effectively optimize and adjust the labor force by measuring the labor force equilibrium degree in the

**Table 4. Algorithm comparison results.**

| Algorithm | Objective function value (day) | Convergence algebra | Success rate (%) |
|---|---|---|---|
| PSO | 256.19 | 120 | 54 |
| Improved PSO | 253.26 | 25 | 98 |

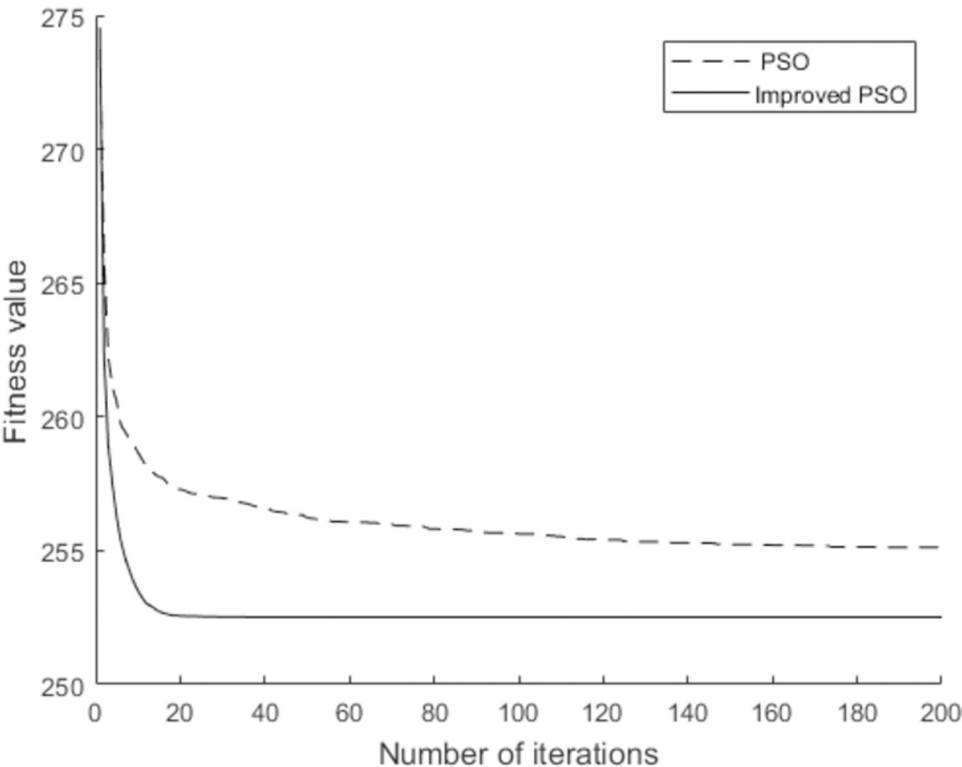

**Fig 8. Comparison of the algorithm evolution process.**

limited working face, reduce the deviation between the labor force distribution and demand, and ensure a labor force balance. In the actual project simulation process, the established labor force equilibrium model aimed at the realization of an extremely short construction period and the model solution algorithm designed based on the PSO algorithm can facilitate the achievement of a labor force balance in each working face, determine the optimal labor force distribution scheme, and finally generate an extremely short construction period of the project of 253.26 days, 27.64% shorter than the contract construction period. In addition, compared to the standard PSO algorithm, the determined extremely short construction period is 256.49 days shorter than that determined with the standard PSO algorithm, and the solution speed is 5.2 times higher. Therefore, the simulation results not only verify the simple operability and practicability of the model but also verify that the designed algorithm (the improved PSO algorithm) achieves a high search accuracy and efficiency in the model solution process.

The results of this study provide a certain theoretical support for managers to realize an extremely short construction period under the condition of resource tolerance. Moreover, against the background of a resource-saving society, it is very important to reduce resource waste and improve resource utilization. However, the model proposed in this study only considers the influencing factor of labor force equilibrium in the determination of an extremely short construction period, and the above examination of the solution method is insufficient. In future research, other factors influencing the realization of an extremely short construction period of engineering projects under the condition of resource tolerance should be comprehensively considered, and other problem solution methods should be further investigated to determine the extremely short construction period of engineering projects under comprehensive effects.

## Author Contributions

**Conceptualization:** Junlong Peng, Mengyao Wang.

**Data curation:** Mengyao Wang, Chao Peng, Ke Hu.

**Funding acquisition:** Junlong Peng.

**Investigation:** Junlong Peng, Mengyao Wang, Chao Peng, Ke Hu.

**Methodology:** Mengyao Wang.

**Resources:** Junlong Peng, Mengyao Wang, Chao Peng, Ke Hu.

**Software:** Mengyao Wang.

**Supervision:** Chao Peng, Ke Hu.

**Validation:** Junlong Peng, Mengyao Wang, Chao Peng, Ke Hu.

**Writing – original draft:** Junlong Peng, Mengyao Wang.

**Writing – review & editing:** Mengyao Wang.

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
