## [Decision Letter · Decision Letter 0]

2 Dec 2021

PONE-D-21-27551Research on extremely short construction period of engineering project based on labor balance under resource tolerancePLOS ONE

Dear Dr. Wang,

Thank you for submitting your manuscript to PLOS ONE. After careful consideration, we feel that it has merit but does not fully meet PLOS ONE’s publication criteria as it currently stands. Therefore, we invite you to submit a revised version of the manuscript that addresses the points raised during the review process.

We look forward to receiving your revised manuscript.

Kind regards,

Ziqiang Zeng, Ph.D.

Academic Editor

PLOS ONE

Journal Requirements:

"This work is supported by the Natural Science Foundation of Hunan Province, China (No: 2021JJ30746)."

"The authors gratefully acknowledge the funding and support provided by the Natural Science Foundation of Hunan Province, China (No: 2021JJ30746)."

4. We suggest you thoroughly copyedit your manuscript for language usage, spelling, and grammar. If you do not know anyone who can help you do this, you may wish to consider employing a professional scientific editing service. 

Additional Editor Comments (if provided):

This paper received three reviewers' comments. While one of them is kind of a positive feedback, the other two reviewers gave several critical comments. Thus, I think a major revision is necessary for this paper and the authors needs to carefully revise this paper according to the review comments, and provide a detailed response in regarding to all the questions the reviewers raised.

Reviewers' comments:

Reviewer's Responses to Questions

**Comments to the Author**

1. Is the manuscript technically sound, and do the data support the conclusions?

Reviewer #1: Yes

Reviewer #2: Yes

Reviewer #3: Partly

2. Has the statistical analysis been performed appropriately and rigorously? 

Reviewer #1: N/A

Reviewer #2: I Don't Know

Reviewer #3: Yes

3. Have the authors made all data underlying the findings in their manuscript fully available?

Reviewer #1: Yes

Reviewer #2: Yes

Reviewer #3: Yes

4. Is the manuscript presented in an intelligible fashion and written in standard English?

Reviewer #1: Yes

Reviewer #2: Yes

Reviewer #3: Yes

5. Review Comments to the Author

Reviewer #1: The paper discusses the extremely short construction period of engineering project based on labor balance under resource tolerance. This topic should arouse the interest of construction project managers. However, several aspects need to be improved or better explained before the paper is approved:

1. As mentioned in line 84 of the introduction, “there are still many factors affecting the realization of extremely short construction period in terms of engineering quantity, management， technology, and environment. This study only focuses on the factor of labor force balance in limited working face.” The author needs to better explain why labor balance can affect the shortening of construction period? Why is labor balance an important factor?

2. The construction of the model is too simple, such as 2.2 research hypothesis. Why does the author make six assumptions? What is the rationality of model assumptions?

3. what is the impact of the six assumptions of the model on the results of the subsequent algorithms? Although there are real cases to prove in the follow-up, we know that the duration of the case is real, and whether the duration obtained by the algorithm deviates due to assumptions?

4. In the six assumptions, there is a sorting error. Please change the following.

5. The paper lists some projects that can be completed in a very short time, but a very important reason for the completion of these projects lies in the different architectural forms, such as the use of assembly and so on. These projects can not well explain the impact of labor balance on shortening the construction period.

6. Is the theme of our article more inclined to traditional architectural forms? If so, please mention it in the hypothesis or final summary.

Reviewer #2: Good work, I wish that we had more examples and comparisons. one Example and one comparison is not enough. There has to be multiples.

The research is sound and the number of pages of the paper are enough to explain the idea.

Reviewer #3: The manuscript describes a technically sound piece of scientific research with data that supports the conclusions. However there are some issues should be considered:

(1) The disequilibrium coefficient in the investigation should be explained clearly, and the scientific and basis of disequilibrium coefficient establishment need to be reasonably explained.

(2) How to explain the effectiveness and scientific of the labor force equilibrium model? It is important in this research.

(3) In order to further verify the superiority of the improved PSO in this paper, we compared the simulation results based on standard PSO and the improved PSO proposed. The analysis is relatively simple. Detailed analysis should be discussed in this section with other methods, and it’s more reasonable.

6. PLOS authors have the option to publish the peer review history of their article (what does this mean?). If published, this will include your full peer review and any attached files.

Reviewer #1: No

Reviewer #2: **Yes: **Dr. Haytham Mahmoud, PE, CGC, CCC

Reviewer #3: No

---

## [Author Response · Author response to Decision Letter 0]

26 Jan 2022

Original Manuscript ID:PONE-D-21-27551

Original Article Title: “Research on extremely short construction period of engineering project based on labor balance under resource tolerance”

To: PLOS ONE Editor

Re: Response to reviewers

Dear Experts and Editor,

Thank you very much for your valuable suggestions on "Research on extremely short construction period of engineering project based on labor balance under resource tolerance". According to the opinions of experts and editor, our team members have modified and improved the corresponding content, please check.

We are uploading (a) our point-by-point response to the comments (below) (response to academic editor and reviewers, the red font indicating modifies), (b) an updated manuscript with yellow highlighting indicating changes (the blue font with strikethrough indicating deletions), (c) a clean updated manuscript without highlights ( main document), (d) supporting Information (English edited manuscript with tracking mark), (e)cover letter (modification of funding statement)

Best regards,

All authors.

Journal Requirements

The manuscript meets PLOS ONE's style requirements.

2. Thank you for stating the following in the Acknowledgments Section of your manuscript: "This work is supported by the Natural Science Foundation of Hunan Province, China (No:2021JJ30746)."

Please remove any funding-related text from the manuscript and let us know how you would like to update your Funding Statement. Currently, your Funding Statement reads as follows: "The authors gratefully acknowledge the funding and support provided by the Natural Science Foundation of Hunan Province, China (No: 2021JJ30746)."

The operation has been carried out as required.

The operation has been carried out as required.

4. In your Data Availability statement, you have not specified where the minimal data set underlying the results described in your manuscript can be found. PLOS defines a study's minimal data set as the underlying data used to reach the conclusions drawn in the manuscript and any additional data required to replicate the reported study findings in their entirety. All PLOS journals require that the minimal data set be made fully available.

"Upon re-submitting your revised manuscript, please upload your study’s minimal underlying data set as either Supporting Information files or to a stable, public repository and include the relevant URLs, DOIs, or accession numbers within your revised cover letter.

The data used to support the findings of this study, including the minimal data set underlying the results described, are within the article.

5. We suggest you thoroughly copyedit your manuscript for language usage, spelling, and grammar. If you do not know anyone who can help you do this, you may wish to consider employing a professional scientific editing service.

Thank you very much for recommending us two professional scientific editing service organizations. We have browsed AJE website and selected Premium Editing. Ruiz P. professionally polished our manuscripts for grammar, phrasing, and punctuation. In addition, many edits were made to further improve the flow and readability of the text., including the following points:

(1)Variety of edits were made to ensure smooth transitions between sentences and to link related thoughts. Sentence flow can be improved by ensuring the appropriate use of conjunctions and introductory words and phrases.

(2)Certain edits were made to remove redundant, repetitive or unnecessary phrasing and to present the information in a more straightforward manner.

(3)Some edits were made to improve conciseness by trimming unnecessary words and streamlining the flow of the manuscript.

Review Comments to the Author

Reviewer#1, Concern # 1: As mentioned in line 84 of the introduction, “there are still many factors affecting the realization of extremely short construction period in terms of engineering quantity, management, technology, and environment. This study only focuses on the factor of labor force balance in limited working face.” The author needs to better explain why labor balance can affect the shortening of construction period? Why is labor balance an important factor?

Author response:

All the above studies have provided an important reference and suggestions for the realization of an extremely short construction period of a given project under resource constraints. In contrast, few scholars have performed research on the achievement of an extremely short construction period under the condition of resource tolerance. However, against the background of COVID-19 and innovation-driven development in the 14th Five-Year Plan [26], it is necessary to thoroughly study the realization of an extremely short construction period of a project from the new perspective of resource tolerance. In the research process, it has been found that even under the condition of resource tolerance, there remain many factors influencing the realization of an extremely short construction period in terms of the engineering quantity [27,28], management [29-31], technology [31,32], and environment [33,34]. This study only focuses on the factor of labor force balance under working face limitations.

Under the condition of resource tolerance, due to the limitation of the working face, we can face the following two situations affecting construction period compression: when the distribution of the labor force in each working face is lower than a certain demand, we cannot increase the construction speed nor minimize the construction period to the highest degree. In addition, many resources (human, financial and material resources) can be wasted. When the distribution of the labor force in each working face is higher than a certain demand, the increase in labor force is not directly proportional to the construction speed. In other words, workers can decrease their work efficiency through working face reduction, thereby affecting the realization of an extremely short construction period. Therefore, under the condition of resource tolerance, it is necessary to perform in-depth research on the realization of an extremely short construction period of a project considering the important influencing factor of labor force balance in the limited working face. We should continuously optimize and adjust the labor force distribution in the limited working face, reduce the deviation between the labor force distribution and demand, balance the labor force distribution and demand, and finally realize an extremely short construction period of the engineering project.

To solve this problem scientifically and effectively, this paper first introduces the stochastic coefficient of labor force equilibrium, which effectively optimizes and adjusts the labor force by measuring the degree of labor force equilibrium in the limited working face. Next, the labor force is balanced by reducing the deviation between the labor force distribution and demand. Then, a labor force equilibrium model with the realization goal of an extremely short construction period is established. Based on a labor force balance in the limited working face, an extremely short construction period of the engineering project can be realized. Finally, the paper improves the standard PSO algorithm from two perspectives: the update equation is rounded to solve practical project problems, and a dynamic inertia weight is adopted to ensure the PSO accuracy and convergence speed. Subsequently, the improved PSO algorithm is employed to solve the research model, and the corresponding extremely short construction period and best labor force distribution scheme are determined. This study can provide theoretical support for project managers to realize an extremely short construction period of engineering projects under the condition of resource tolerance. 

References:

[26] Xi J. P. Statement on the proposal of the Central Committee of the CPC on formulating he 14th Five-Year Plan (2021-2025) for National Economic and Social Development and the Long-Range Objectives Through the Year 2035 [N]. people's daily, 2020-10-29.

[27] Bayram S. Duration Prediction Models for Construction Projects: In Terms of Cost or Physical Characteristics? [J]. KSCE journal of civil engineering, 2017, 21(6):2049-2060.

[28] Khatib B. A, Poh Y. S, El-Shafie A. Delay Factors Management and Ranking for Reconstruction and Rehabilitation Projects Based on the Relative Importance Index (RII)[J]. Sustainability,2020,12(15):

[29] Chan D, Kumaraswamy M. M. Compressing construction durations: lessons learned from Hong Kong building projects[J]. International Journal of Project Management, 2002, 20(1):23-35.

[30] Doloi H, Sawhney A, Iyer K. C, Rentala S. Analyzing factors affecting delays in Indian construction projects[J]. International Journal of Project Management, 2012, 30(4):479-489.

[31] Suresh V, Patel A, Ramachandran B. Attitude toward COVID-19 vaccination: A cross-sectional study on healthcare professionals. [J]. Indian journal of pharmacology,2021,53(3):

[32] Jin R, Han S, Hyun C. T, Cha Y. Application of Case-Based Reasoning for Estimating Preliminary Duration of Building Projects[J]. Journal of Construction Engineering and Management,2015:

[33] Aibinu A. A, Odeyinka H A. Construction Delays and Their Causative Factors in Nigeria [J]. Journal of Construction Engineering and Management, 2006, 132(7).

[34] Alsuliman J. A. Causes of delay in Saudi public construction projects [J]. Alexandria Engineering Journal, 2019, 58(2):

Reviewer#1, Concern # 2: The construction of the model is too simple, such as 2.2 research hypothesis. Why does the author make six assumptions? What is the rationality of model assumptions?

Author response:

After careful inspection, there are five assumptions listed in this paper. The reasons and rationality of setting assumptions are explained as follows: 

Explanation for hypothesis (1):"Quality, progress and cost" are the three major objectives of the engineering project. However, under the condition of a great quantity of human, financial and material resources gathering, that is, resource tolerance, project managers usually need to solve the "quality" and "speed" trade-off problems [Gao, 2012].Under the condition of resource tolerance, the main content of this paper is to focus on the influencing factor of labor force balance on the limited working face , so as to realize the extremely short working period of the project. The purpose of the hypothesis (1) is to show that the quality problem of the project is guaranteed while obtaining an extremely short construction period.

Explanation for hypothesis (2): In this study, the duration of each activity is not rounded in order to obtain a more accurate extremely short working period and better reflect the meaning of "extremely short".

Explanation for hypothesis (3): In the process of model construction, the engineering quantity of each activity is indicated as , the duration of each activity can be calculated as follows:

 （7）

As we can see from Eq (7), the value of directly affects the value of , and then affects the determination of the construction period of the project. The purpose of hypothesis (3) is to facilitate the subsequent research in this paper and ensure the significance of the research. It does not consider the situation that some activities in the project change their quantities due to emergencies, which makes the construction period an uncertain value.

Explanation for hypothesis (4): The purpose of this hypothesis is to explain that under the condition of resource tolerance, the labor force with different skills required in each working face is sufficient. There is no situation that one activity occupies the labor force of another activity, so that another activity is delayed or cannot be carried out according to the normal plan. On this basis, the problem of labor balance on the limited working face is solved to pursue and realize the extremely short construction period of the project.

Explanation for hypothesis (5): When the labor force is unbalanced, we cannot define the impact on the construction period when the labor force distribution on each working face is greater than or less than the same unit of labor force demand. To avoid disputes, we make this hypothesis.

In order to more accurately express the meaning and rationality of the assumptions, we have made language modifications to the five assumptions, as follows:

(1) Under the condition of resource tolerance, this paper achieves an extremely short construction period with quality assurance.

(2) The duration of each activity is not rounded to preserve the accuracy of the determination of an extremely short construction period.

(3) The operation process of each activity cannot be interrupted, and the quantities of each activity remain fixed.

(4) Under the condition of resource tolerance, the labor force distribution in the working face of each activity is independent, and there occurs no delay or failure to conduct an activity according to the normal plan due to an insufficient labor force.

(5) The impact on the construction period is the same when the labor force distribution in the working face of each activity is higher than or lower than the same unit of the labor force demand.

References:

Gao Jun. Research on paradigm and strategy of rural housing construction based on Wenchuan earthquake reconstruction [D], Zhejiang University, 2012, doctor.

Reviewer#1, Concern # 3: what is the impact of the six assumptions of the model on the results of the subsequent algorithms? Although there are real cases to prove in the follow-up, we know that the duration of the case is real, and whether the duration obtained by the algorithm deviates due to assumptions?

Author response:

The duration obtained by the algorithm will deviate due to assumptions. Especially hypothesis (2) and hypothesis (3), if there are no these two assumptions, it will directly affect the duration of each activity, and then affect the accuracy of the solution results of the extremely short construction period. The specific explanation has been answered in concern # 2.

Reviewer#1, Concern # 4: In the six assumptions, there is a sorting error. Please change the following.

Author response:

To facilitate analysis, the following hypotheses are established:

(1) Under the condition of resource tolerance, this paper achieves an extremely short construction period with quality assurance.

(2) The duration of each activity is not rounded to preserve the accuracy of the determination of an extremely short construction period.

(3) The operation process of each activity cannot be interrupted, and the quantities of each activity remain fixed.

(4) Under the condition of resource tolerance, the labor force distribution in the working face of each activity is independent, and there occurs no delay or failure to conduct an activity according to the normal plan due to an insufficient labor force.

(5) The impact on the construction period is the same when the labor force distribution in the working face of each activity is higher than or lower than the same unit of the labor force demand. 

Reviewer#1, Concern # 5: The paper lists some projects that can be completed in a very short time, but a very important reason for the completion of these projects lies in the different architectural forms, such as the use of assembly and so on. These projects cannot well explain the impact of labor balance on shortening the construction period.

Author response:

Buildings using assembly are called prefabricated buildings or modular buildings [Hong, 2018]. Compared with traditional buildings (cast-in-situ concrete buildings), it can minimize the construction time and complete the project in a very short time. [Hu, 2019 and El Abidi, 2019] However, none of the current engineering projects is fully automated. Even buildings with assembly and other rapid construction technologies need the participation of workers. Therefore, under the condition of resource tolerance, both traditional and assembly projects will face the problem of labor force balance on the limited working face, so that the construction period cannot be further shorten to the limit.

References:

Hong J, Shen G Q, Li Z.B, Zhang W. Barriers to promoting prefabricated construction in China: A cost-benefit analysis [J]. Journal of Cleaner Production. 2018, 172:649-660.

Hu X, Chong H Y, Wang X. Understanding stakeholders in Off-Site manufacturing: a literature review[J]. Journal of Construction Engineering and Management. 2019, 145(8):1-15.

El-Abidi K, Ofori G, Zakaria S, Aziz A. Using prefabricated building to address housing needs in Libya: a study based on local expert perspectives[J]. Arabian Journal for Science and Engineering. 2019,44(10):8289-8304.

Reviewer#1, Concern # 6: Is the theme of our article more inclined to traditional architectural forms? If so, please mention it in the hypothesis or final summary.

Author response:

This paper is not only applicable to traditional buildings, but also to other buildings using advanced technology. The specific explanation has been answered in concern # 5.

Reviewer#3, Concern # 1: The disequilibrium coefficient in the investigation should be explained clearly, and the scientific and basis of disequilibrium coefficient establishment need to be reasonably explained.

Author response:

The disequilibrium coefficient proposed in this paper refers to the following two literatures [Cai, 2019 and Jia, 2011] to measure the imbalance of resource consumption. Because it is inconsistent with the scope and specific problems of this study, we rename the "disequilibrium coefficient" as "stochastic coefficient of labor force equilibrium" and revise it in the whole paper. This not only ensures the preciseness of the article and avoids disputes, but also can be regarded as a small innovation point of this study. The specific revises are as follows:

3.1 Stochastic coefficient of labor force equilibrium 

In this study, the goal of realizing an extremely short construction period of the project is reached under the premise of a labor force balance in the limited working face of each activity. Hence, to measure the degree of labor balance in the working face, we introduced the stochastic coefficient of labor force equilibrium . Notably, the imbalance in the labor force can be divided into two cases in this paper: and . Therefore, the expression of the stochastic coefficient of labor force equilibrium () is as follows: 

(1)

Where denotes the stochastic coefficient of labor force equilibrium in the working face of each activity. When the value of approaches 1, the labor force becomes increasingly balanced. For (), the labor force is completely balanced and reaches the ideal state. is a constant greater than 1 and represents the maximum acceptable value of the stochastic coefficient of labor force equilibrium. In the limited working face, given the safe distance and working efficiency, .

References:

Cai Qianfen, Wang Junwu Improvement of minimum variance method for labor resource optimization of construction project [J] Statistics and decision making, 2019,35 (13): 181-184.

Jia B. P, Liu L. L, Lu Q. Construction organization and management of construction engineering [M] Xi'an: Xi'an Jiaotong University Press, 2011.

Reviewer#3, Concern # 2: How to explain the effectiveness and scientific of the labor force equilibrium model? It is important in this research.

Author response:

this paper will finally realize the extremely short construction period of the project by continuously optimizing and adjusting the labor force distribution on the limited working face, reducing the deviation between the distribution and a certain demand of labor force, and making the labor force tend to be balanced. In Section 5.2 simulation result, we illustrate (verify) the scientific effectiveness of the labor force equilibrium model from the solution results and the reliability of the result data. The specific contents are as follows:

Table 3 reveals that the extremely short construction duration of the project reaches 253.26 days. Compared to the contract period, the construction period is 27.64% shorter. Moreover, the specific duration of each activity is provided in Table 3 and intuitively shown in Fig. 4. Consequently, the critical path of this project is →→→→→→→→→→. Furthermore, a bar chart of the schedule corresponding to the obtained extremely short construction period of the project was generated, as shown in Fig. 5.

Table 3 and Fig. 6 show that the actual labor force distribution of each activity was obtained. Actually, the result represents the best labor force distribution scheme of the project. Fig. 7 shows the distribution of the stochastic coefficient of labor force equilibrium () for each activity. Except for the activities involving human–machine cooperation, the equilibrium deviation is no greater than 0.100, indicating that an extremely short construction period is realized based on the balance among the various working labor forces. Thus, the obtained scheme achieves a suitable reliability. Moreover, the solution process gradually converges in this study. After approximately 25 generations, convergence is accomplished to yield the optimal solution, which verifies the feasibility of the model and algorithm to solve practical problems of engineering projects (the convergence process is described in the next section).

Reviewer#3, Concern # 3: The choice of the method, particularly its advantages over other existing approaches should be highlighted. What I’m missing here is the comparison with other algorithms. This would not only show the advantages, but also validate the results. If it is too much to include such a comparison, at least the other approaches should be mentioned.

Author response:

Thank you very much for your comments. In Section 4, we described the advantages of PSO and its applicability to this research problem. In addition, in order to get better research results, the standard PSO was adjusted in 4.3 and 4.4. Before case simulation and algorithm performance comparison, the performance of improved PSO proposed was compared and analyzed by using the test function in 4.4, which is consistent with the algorithm performance comparison in 5.3. For the standard PSO , this not only shows the superiority of the improved PSO we proposed, but also verifies the reliable applicability and superiority of the improved PSO in solving the practical problems in this study.

As the experts said, we regret to ignore the discussion combined with other methods to make it more reasonable. There are few relevant documents involved in realizing the extremely short construction period of engineering projects under resource tolerance. If detailed analysis should be discussed in this section with other methods, it will take a lot of time to learn other methods and their code programming. More importantly, it is necessary to explore the applicability, effectiveness, accuracy and efficiency of other methods to this research problem. I'm afraid we can't make a scientific and reasonable analysis before submitting the revision. Also, the important content of this study, such as the expert pointed out in concern # 2: the validation of the effectiveness and scientificity of the labor force balance model. Therefore, in order to ensure the scientific preciseness of this research, we will revise the conclusion appropriately: increasing the data description to highlight the focus of this study; supplementing the limitations of the article to point out the slight weakness of the analysis part of this research method and providing the research direction in the future.

The specific revises are as follows:

Under the condition of resource tolerance, based on a labor force balance in the limited working face, an extremely short construction period of the project can be realized. This study demonstrates that the stochastic coefficient of labor force equilibrium introduced can effectively optimize and adjust the labor force by measuring the labor force equilibrium degree in the limited working face, reduce the deviation between the labor force distribution and demand, and ensure a labor force balance. In the actual project simulation process, the established labor force equilibrium model aimed at the realization of an extremely short construction period and the model solution algorithm designed based on the PSO algorithm can facilitate the achievement of a labor force balance in each working face, determine the optimal labor force distribution scheme, and finally generate an extremely short construction period of the project of 253.26 days, 27.64% shorter than the contract construction period. In addition, compared to the standard PSO algorithm, the determined extremely short construction period is 256.49 days shorter than that determined with the standard PSO algorithm, and the solution speed is 5.2 times higher. Therefore, the simulation results not only verify the simple operability and practicability of the model but also verify that the designed algorithm (the improved PSO algorithm) achieves a high search accuracy and efficiency in the model solution process.

The results of this study provide a certain theoretical support for managers to realize an extremely short construction period under the condition of resource tolerance. Moreover, against the background of a resource-saving society, it is very important to reduce resource waste and improve resource utilization. However, the model proposed in this study only considers the influencing factor of labor force equilibrium in the determination of an extremely short construction period, and the above examination of the solution method is insufficient. In future research, other factors influencing the realization of an extremely short construction period of engineering projects under the condition of resource tolerance should be comprehensively considered, and other problem solution methods should be further investigated to determine the extremely short construction period of engineering projects under comprehensive effects.

---

## [Decision Letter · Decision Letter 1]

14 Mar 2022

Research on extremely short construction period of engineering project based on labor balance under resource tolerance

PONE-D-21-27551R1

Dear Dr. Wang,

We’re pleased to inform you that your manuscript has been judged scientifically suitable for publication and will be formally accepted for publication once it meets all outstanding technical requirements.

Kind regards,

Ziqiang Zeng, Ph.D.

Academic Editor

PLOS ONE

Additional Editor Comments (optional):

Based on the reviewers' suggestions, this paper is ready for publication.

Reviewers' comments:

Reviewer's Responses to Questions

**Comments to the Author**

1. If the authors have adequately addressed your comments raised in a previous round of review and you feel that this manuscript is now acceptable for publication, you may indicate that here to bypass the “Comments to the Author” section, enter your conflict of interest statement in the “Confidential to Editor” section, and submit your "Accept" recommendation.

Reviewer #1: (No Response)

Reviewer #2: All comments have been addressed

Reviewer #3: All comments have been addressed

2. Is the manuscript technically sound, and do the data support the conclusions?

Reviewer #1: Yes

Reviewer #2: Yes

Reviewer #3: Yes

3. Has the statistical analysis been performed appropriately and rigorously? 

Reviewer #1: Yes

Reviewer #2: Yes

Reviewer #3: Yes

4. Have the authors made all data underlying the findings in their manuscript fully available?

Reviewer #1: Yes

Reviewer #2: Yes

Reviewer #3: Yes

5. Is the manuscript presented in an intelligible fashion and written in standard English?

Reviewer #1: Yes

Reviewer #2: Yes

Reviewer #3: Yes

6. Review Comments to the Author

Reviewer #1: (No Response)

Reviewer #2: I enjoyed the paper this time around. All the data was clear and graphs in place thank you for the work.

Reviewer #3: Under the condition of resource tolerance, engineering construction projects face the problem of labor force balance in the working face. Notably, a deviation occurs between the distribution and certain demand of the labor force in the limited working face, which affects the realization of an extremely short construction period. The auhors revised the paper according to the reviewers. My comments have been well addressed, and I think is OK to accept it.

7. PLOS authors have the option to publish the peer review history of their article (what does this mean?). If published, this will include your full peer review and any attached files.

Reviewer #1: No

Reviewer #2: **Yes: **Dr. Haytham Mahmoud, PE, CGC, CCC

Reviewer #3: No

---

## [Editor Report · Acceptance letter]

22 Mar 2022

PONE-D-21-27551R1 

Research on extremely short construction period of engineering project based on labor balance under resource tolerance 

Dear Dr. Wang:

I'm pleased to inform you that your manuscript has been deemed suitable for publication in PLOS ONE. Congratulations! Your manuscript is now with our production department. 

Kind regards, 

on behalf of

Dr. Ziqiang Zeng 

Academic Editor

PLOS ONE